# Integrated Management of Childhood Illnesses (IMCI): a mixed-methods study on implementation, knowledge and resource availability in Malawi

Kim Kilov,[1] Helena Hildenwall,[1,2,3] Albert Dube,[4] Beatiwel Zadutsa,[5] Lumbani Banda,[5] Josephine Langton,[6] Nicola Desmond,[7] Norman Lufesi,[8] Charles Makwenda,[5] Carina King  [1,9]

For numbered affiliations see end of article.

**Correspondence to**
Dr Carina King; carina.king@ki.se

## ABSTRACT

**Background**  The introduction of the WHO's Integrated Management of Childhood Illnesses (IMCI) guidelines in the mid-1990s contributed to global reductions in under-five mortality. However, issues in quality of care have been reported. We aimed to determine resource availability and healthcare worker knowledge of IMCI guidelines in two districts in Malawi.

**Methods**  We conducted a mixed-methods study, including health facility audits to record availability and functionality of essential IMCI equipment and availability of IMCI drugs, healthcare provider survey and focus group discussions (FGDs) with facility staff. The study was conducted between January and April 2019 in Mchinji (central region) and Zomba (southern region) districts. Quantitative data were described using proportions and $\chi^2$ tests; linear regression was conducted to explore factors associated with IMCI knowledge. Qualitative data were analysed using a pragmatic framework approach. Qualitative and quantitative data were analysed and presented separately.

**Results**  Forty-seven health facilities and 531 healthcare workers were included. Lumefantrine-Artemether and cotrimoxazole were the most available drugs (98% and 96%); while amoxicillin tablets and salbutamol nebuliser solution were the least available (28% and 36%). Respiratory rate timers were the least available piece of equipment, with only 8 (17%) facilities having a functional device. The mean IMCI knowledge score was 3.96 out of 10, and there was a statistically significant association between knowledge and having received refresher training (coeff: 0.42; 95% CI 0.01 to 0.82). Four themes were identified in the FGDs: IMCI implementation and practice, barriers to IMCI, benefits of IMCI and sustainability.

**Conclusion**  We found key gaps in IMCI implementation; however, these were not homogenous across facilities, suggesting opportunities to learn from locally adapted IMCI best practices. Improving on-going mentorship, training and supervision should be explored to improve quality of care, and programming which moves away from vertical financing with short-term support, to a more holistic approach with embedded sustainability may address the balance of resources for different conditions.

## What is known about the subject?

► WHO's integrated management of childhood illness (IMCI) guidelines have been important in reducing child deaths in low-resource settings.
► However, quality of care issues in IMCI implementation have been noted, including lack of resources, drugs and trained staff.
► Training in IMCI is variable, and recommendations for and evidence on the impact of refresher trainings is limited.

## What this study adds?

► In two districts in Malawi, we found key IMCI implementation gaps; these were not homogenous across facilities suggesting opportunities to learn from locally adapted best practice.
► Training alone was not independently associated with improved IMCI knowledge, but having had refresher training was.
► Resource gaps were more prominent for respiratory and diarrhoeal case management than malaria, highlighting issues in vertical financing.

## INTRODUCTION

Despite a reduction in under-five deaths globally from 12.5 million in 1990 to 5.3 million in 2018, progress has been uneven.[1] The highest burden is seen in sub-Saharan Africa, with considerable national and sub-national variation.[1] A key strategy in the success to date was the introduction of the WHOs Integrated Management of Childhood Illness (IMCI) guidelines.[2] At the time, the major causes of child mortality were pneumonia, malaria, measles, malnutrition and diarrhoea.[2] More recently this has shifted to neonatal complications,[1] while pneumonia remains the leading infectious cause of mortality in children under-five.[3]

IMCI was intended for countries with under-five mortality rates higher than 40 per 1000 live births, and focuses on three components: improving case management skills, strengthening health systems and improving community health practices.[4 5] While IMCI has been partially or fully adopted by 100 countries,[6 7] a survey from 2016 reported only 44 countries were considered to be fully implementing it.[2] Further, IMCI coverage was lowest in countries with the highest mortality rates; of the 26 countries to achieve the Millennium Development Goal 4 (MDG4)—reducing under-five mortality by two-thirds, 20 had fully implemented IMCI.[2]

Key strengths of IMCI are the holistic approach, rational use of medications and improved quality and efficiency of health service provision.[2 7] However, implementation barriers and inconsistencies have been reported. For example, the WHO guidance for training contains seven modules taught over 11 days, but local adaptations have led to considerable differences, with examples of both shorter and distance learning approaches.[8 9] Currently there is a lack of standardised guidance on the frequency, content, pedagogical approach and expected learning outcomes for refresher courses.

Malawi was an early adopter of IMCI, first implementing it in 2000,[10] and was one of only 10 low-income countries in sub-Saharan Africa to achieve MDG4.[11] This success has been attributed to proactive policies, scale-up and introduction of vaccinations, coverage of insecticide treated bed nets, IMCI and integrated community case management.[10] However, Malawi is not currently on track to achieve the target set out in Sustainable Development Goal 3.2, and faces considerable challenges with an under-resourced health sector with workforce shortages.[12] Previous studies have reported suboptimal IMCI implementation for pneumonia, with issues in clinical assessments, quality of diagnosis and antibiotic prescription.[13 14]

Given changes in causes of under-five mortality, varied successes in IMCI implementation and ongoing challenges of under-resourced health systems, it is important to assess if IMCI is effectively supported and implemented. We aimed to describe current IMCI implementation at primary care in Malawi, and determine whether there were sufficient resources and trained staff available to assess and manage children under-five according to these guidelines.

## METHODS

We conducted a concurrent mixed-methods study, covering all dispensaries, health centres, rural hospitals and Christian Health Association of Malawi (CHAM) hospitals in Zomba and Mchinji districts. Data were collected between January and April 2019. The study included facility audits to assess resource availability, healthcare provider surveys to assess training and knowledge, and focus group discussions (FGDs) with healthcare providers and facility managers to explore current implementation. Qualitative and quantitative data were analysed and are presented separately.

### Setting

Mchinji district is in Malawi's central region, with an under-five population of approximately 90 000 and an under-five mortality rate of 123/1000 live births in the 2015–2016 Demographic Health Survey. Zomba is located in the southern region, with an under-five population of 120 000 and mortality of 54/1000 live births.[15 16] Zomba has a larger urban population and has historically received both higher per capita domestic and external funding,[15 17] while Mchinji has 84% of the population living rurally.[15] The health system is made up of three levels.[18] Health centres, dispensaries and rural clinics deliver primary care and are linked to community care via Health Surveillance Assistants (HSAs).[19] Secondary care is delivered at district hospitals and regional referral hospitals provide tertiary care. A summary of healthcare workers and their roles are presented in online supplemental appendix 1. Government care is free, and costs of services delivered by CHAM facilities are subsidised through service level agreements resulting in small out of pocket payments.

### Sampling

All dispensaries, health centres, rural and CHAM hospitals (ie, frontline facilities) were eligible to be audited. A convenience sample of staff were recruited for the survey, including staff who interact with paediatric patients and were present at the facility at the time of data collection. Qualitative participants were selected using purposive sampling, aiming to conduct three FGDs in each district. Participants were invited to participate by phone by a member of study staff, with permission from the District Health Management Team and senior HSAs. The groups targeted included: (1) HSAs and attendants at health centres; (2) medical assistants and nurses at health centres; (3) clinical officers and medical assistants at rural hospitals. We invited up to 10 participants per group.

### Quantitative data collection

The audit assessed the number of healthcare workers, availability and functionality of essential IMCI equipment (thermometer, respiratory rate timer, mid-upper arm circumference (MUAC) tape, scale, nebuliser) and availability of IMCI drugs (antibiotics, antimalarials, oral rehydration solution and salbutamol) used for outpatient case management and prereferral treatment, and whether they were in date. Data on drugs and equipment used for complex emergency cases are not presented. The structured healthcare worker survey included questions on: cadre, previous IMCI training, refresher courses, years since training and refresher training received and knowledge of the 2014 IMCI guidelines. The knowledge questions were adapted from an IMCI computer-based training course evaluation delivered by USAID in Kenya[20]; this was selected due to its short length covering different

elements of IMCI knowledge, and the questions should not have been previously seen by participants in our setting.

Data were collected by trained study staff who visited each facility at a prearranged time to conduct the audit and surveys. The study staff consisted of clinical officers and monitoring and evaluation officers. The data were collected with support from facility in-charge, pharmacy and health management information staff and included visual inspection of drug stocks to check quantities and expiry dates, equipment functionality and closed questions. Visual inspection by the study staff member was conducted to reduce potential recall and social desirability biases. Data were entered into android tablets using Open Data Kit Collect, with in-built cleaning and skip-pattern rules to promote data quality. The survey was self-completed on an android tablet for healthcare workers who were familiar with smart phone technology. If unfamiliar, the survey was interviewer administered.

## Qualitative data collection

FGDs were held at healthcare facilities and led by a male Malawian researcher (AD) with experience of qualitative research, supported by a clinical officer to provide IMCI specific knowledge. Discussions were open to be conducted in Chichewa or English, depending on the content and preference of participants. Interviews and discussions were audio-recorded, transcribed and where necessary translated into English. Participants were reimbursed for their travel expenses and provided with refreshments.

## Analysis

The audit data were described using proportions, and compared between facility types and districts using $\chi^2$ tests. Healthcare worker knowledge was summarised into a score with a maximum of 10 points. Mean scores were compared for the following independent variables: IMCI training received, years since training, refresher training received, years since refresher training, cadre, facility type and district. Multivariable linear regression analysis was conducted to assess the association between IMCI knowledge scores and training. The following confounders were considered: district (given different funding levels), facility type, health worker qualification and refresher course. Quantitative analysis was performed using Stata IC V.16.0.

FGDs were analysed using a pragmatic framework approach, with predefined themes based on the topic guide[21] (online supplemental appendix 2). Emergent themes were coded during the analysis. Discrepancies were agreed through discussion, and the interpretation and conclusions shared and discussed with the wider study team. Coding was done by HH, with a subset double coded by CK. These codes were then discussed, refined and organised into themes, which were checked by AD for consistency with his interpretation and the local context.

### Table 1 Health facility and healthcare worker cadre inclusion, by district

| Facilities | Total | Mchinji | Zomba |
|---|---|---|---|
| Dispensary | 7 | 4 | 3 |
| Health centre | 32 | 8 | 24 |
| Rural hospital | 3 | 1 | 2 |
| CHAM hospital | 5 | 3 | 2 |
| Total | 47 | 16 | 31 |
| **Healthcare worker survey respondents** | | | |
| Clinical officer | 17 (3%) | 6 (3%) | 11 (3%) |
| Medical assistant | 47 (9%) | 15 (8%) | 32 (9%) |
| Nurse/midwife | 105 (20%) | 32 (17%) | 73 (22%) |
| HSA | 190 (36%) | 69 (36%) | 121 (36%) |
| Hospital attendant | 172 (32%) | 71 (37%) | 101 (30%) |
| Total | 531 | 193 | 338 |

CHAM, Christian Health Association of Malawi; HSA, Health Surveillance Assistant.

## Patient and public involvement

Patients and public were not involved in the design or execution of the study. Prior to starting, the protocol was presented to both District Health Management and District Executive Committees (which include local government and civil society representation) in both Zomba and Mchinji. Minor edits to the study plan were instituted following feedback and questions raised in these meetings.

## RESULTS

The total 47 health facilities were audited, with 16 in Mchinji and 31 in Zomba (table 1), and 44% (n=531/1197) of clinical staff employed across these facilities were surveyed. All six planned FGDs were completed, with three in each district.

### Facility audits

Availability of seven IMCI drugs was assessed (table 2). Lumefantrine-Artemether and cotrimoxazole were the most available (98% and 96%); while amoxicillin tablets and salbutamol nebuliser solution were the least available (28% and 36%). When available, the majority of drugs were found to be in date. Respiratory rate timers were the least available piece of equipment, with only 8 (17%) facilities across both districts having a functional device. This was followed by pulse oximeters (30%) and micronebulisers (32%, table 3). MUAC tapes, scales and malaria rapid diagnostic tests (mRDTs) were almost universally available. Health centres had the highest proportion of non-functional equipment.

### Healthcare worker survey

Overall 42% (n=222/531) of the survey respondents reported having IMCI training, with no difference by district (p value=0.900). Of these, 38% had also received refresher training. The most common group to report

**Table 2** Drug availability and proportion of drugs in date by facility type

| | CHAM hospital (n=5) | Rural hospital (n=3) | Health centre (n=32) | Dispensary (n=7) |
|---|---|---|---|---|
| Cotrimoxazole tablet | 4 (80%) | 3 (100%) | 32 (100%) | 6 (86%) |
| Amoxicillin tablets* | 5 (100%) | 1 (33%) | 6 (19%) | 1 (14%) |
| Amoxicillin syrup | 4 (80%) | 0 | 20 (63%) | 4 (57%) |
| Lumefantrine-Artemether | 5 (100%) | 3 (100%) | 32 (100%) | 6 (86%) |
| Rectal artesunate | 1 (20%) | 1 (33%) | 19 (59%) | 1 (14%) |
| Oral rehydration salts | 5 (100%) | 0 | 16 (50%) | 3 (43%) |
| Salbutamol tablets | 5 (100%) | 2 (67%) | 15 (47%) | 3 (43%) |
| Salbutamol nebuliser solution | 4 (80%) | 1 (33%) | 5 (16%) | 7 (100%) |
| Aminophylline injection† | 2 (67%) | 1 (100%) | 3 (38%) | 3 (75%) |

The following drugs were found not to be in date: 20% of oral rehydration salts at CHAM facilities; 5% of amoxicillin syrup in health centres and 25% at dispensaries; 33% aminophylline at health centres.
*Availability of amoxicillin syrup and/or tables: 5 (100%) of CHAM facilities; 1 (33%) rural hospital; 22 (69%) health centres; 4 (57%) dispensaries.
†Data only collected for Mchinji district (n=16).
CHAM, Christian Health Association of Malawi.

training were HSAs (78%), while clinical officers, medical assistants and nurses all reported similar levels (45%), and only one attendant reported training. Training was most frequently received from non-governmental organisations (NGOs) (50%) followed by Ministry of Health (36%) and during qualification training (14%); refresher trainings were delivered by NGOs (64%) and Ministry of Health (36%). The median time since initial training was 7 years (95% CI 6 8; range: 0–21).

The mean knowledge score, of a possible 10 points, was 3.96 (95% CI 3.83 to 4.08; range: 0–8), with 65% scoring less than 5 (online supplemental appendix 3). Questions on referral decision using clinical scenarios had both the most correct responses (84%—severe dehydration with another sign of severity), and most incorrect (13%—severe dehydration without any other urgent signs). The multivariable regression found a statistically significant association between IMCI knowledge score and having received refresher training (coeff: 0.42; 95% CI 0.01 to 0.82)—table 4. However, training alone was not significantly associated with IMCI knowledge. Being an HSA or hospital attendant were both associated with a lower score compared with clinical officers (coeff: −0.82; 95% CI −1.50 to −0.14 and coeff: −1.61; 95% CI −2.29 to −0.92). Among those with training, there was a weak but statistically significant positive correlation between knowledge and years since training (correlation coeff: 0.213; p value: 0.001).

### Focus group discussions

The qualitative data are presented under the following themes: IMCI implementation and practice, barriers to IMCI effectiveness, benefits of IMCI and sustainability.

### IMCI implementation and practice

This theme includes healthcare workers understanding of IMCI, real-world adaptations, self-perceived quality of implementation, and training received. Overall IMCI was described as a system of assessment, a holistic approach to care for children and as a way to classify illness severity.

Across the discussions it was apparent that there were inconsistencies in the implementation of IMCI, but groups generally described a process of weighing, initial clinical assessment, tests and then assigning a diagnosis. One group from Mchinji specifically talked of how a new electronic patient record system had resulted in more structure.

Healthcare workers discussed doing their best to deliver IMCI, in spite of challenges. Although also acknowledged that they do not always follow the guidelines, whether this was intentional was not clear:

> I guess some people are not using the IMCI approach so it is hard to assess properly. Because they are straight from school sometimes you need updates in order to know when everything is working properly and to give an alarm to the people who are using it (Zomba, FGD 1)

In the groups, there was a mix between those who reported training and those who had not received any. Participants in one group described how staff had also received different types of training, depending on who delivered it:

> There were some people who were trained by the Ministry of health, while others were retained by PSI [an international NGO …] it was a week-long training. But those who were trained by other organisations, the training lasted for three weeks. (Zomba, FGD 2)

### Barriers to IMCI effectiveness

Three main barriers were discussed: community barriers, lack of adequate resources and lack of trained staff capacity. The issue of caregivers presenting late was a key example raised by several groups as a barrier to effective IMCI implementation. This resulted in tensions

**Table 3** Equipment availability by facility type and proportion of functional equipment

| | CHAM hospital (n=5) | | Rural hospital (n=3) | | Health centre (n=32) | | Dispensary (n=7) | |
|---|---|---|---|---|---|---|---|---|
| | Available (n (%)) | Functional (%) | Available (n (%)) | Functional (%) | Available (n (%)) | Functional (%) | Available (n (%)) | Functional (%) |
| Thermometer | 5 (100) | 100 | 3 (100) | 100 | 25 (78) | 96 | 7 (100) | 71 |
| Respiratory rate timer | 1 (20) | 100 | 0 | – | 6 (19) | 67 | 2 | 100 |
| MUAC tape | 5 (100) | 100 | 3 (100) | 100 | 32 (100) | 100 | 7 (100) | 100 |
| Weighing scale | 5 (100) | 100 | 3 (100) | 100 | 32 (100) | 100 | 7 (100) | 100 |
| Micro-nebuliser | 3 (60) | 100 | 1 (33) | 100 | 9 (28) | 78 | 2 | 100 |
| Malaria RDTs | 5 (100) | 100 | 3 (100) | 100 | 31 (97) | 100 | 7 (100) | 100 |
| Pulse oximeter* | 3 (60) | 100 | 1 (33) | 100 | 10 (31) | 70 | 0 | – |

*Pulse oximetry is recommended but not essential according to Integrated Management of Childhood Illnesses guidelines.
CHAM, Christian Health Association of Malawi; MUAC, mid-upper arm circumference; RDT, rapid diagnostic test.

around communication and trust between caregivers and providers, and participants were aware of the compounding effects of high patient loads, stock-outs and not having 24-hour staffing.

> When we ask her the time the child started getting sick she tells a lie by saying they started getting ill today. If we argue [with] her response, you find that sometimes neighbours will confront her by saying the child started suffering sometime back only that she was busy running her business. That's the challenge we encounter sometimes. (Zomba, FDG 3)

While the presence of community structures to support the facilities were described, they noted that they did not always work together effectively. In terms of resources, this was raised as a major challenge for some, but not others who described well-functioning supply chain management for essential medications, such as antibiotics. Various scenarios of lacking guidelines, medications, diagnostics and equipment were given to highlight how these limit the ability to provide care for children.

> Sometimes you can find that the child has high fever and there is a need for the child to be given [antimalarials], but before that you have to confirm with a malaria test, and if you don't have the test kit it becomes a big challenge (Zomba, FDG 2)

> The guidelines are not there. Even if you are not trained you can use the guidelines to tell you what to do, but the guidelines are not there (Zomba, FDG 1)

Multiple dimensions of staffing were raised, from high workloads restricting the amount of time that could be spent conducting an IMCI assessment, not all staff being trained, and lacking supportive supervision. One discussion with providers from a CHAM hospital however noted that they had no problems with staffing unless there were absences due to events such as funerals.

## Benefits of IMCI

Benefits of IMCI included better clinical outcomes and better ways of working. Generally, many of the participants acknowledged that IMCI results in improved outcomes for children, and that this also translates into care and care-seeking practices among the community.

> It helps mothers to have additional knowledge of caring [for their] children, through what she is advised at hospital by health workers (Zomba, FDG 2)

IMCI was described as reducing the workload for hospitals, as frontline facility staff were able to treat conditions such as malaria. The structured and holistic approach was valued by participants in both districts, for example:

**Table 4**  IMCI knowledge score description and multivariable linear regression analysis

| Variables | Mean score (n=531) | | | | Coefficient (95% CI) | P value |
|---|---|---|---|---|---|---|
| | N | Mean | Mchinji (n=193) | Zomba (n=338) | | |
| **Training** | | | | | | |
| No training | 309 | 3.69 | 3.54 | 3.77 | Ref | |
| IMCI training only | 137 | 4.31 | 4.40 | 4.26 | 0.083 (−0.238 to 0.405) | 0.611 |
| IMCI training+refresher | 85 | 4.40 | 4.26 | 4.50 | 0.417 (0.012 to 0.821) | 0.043 |
| **Cadre** | | | | | | |
| Clinical officer | 17 | 4.76 | 4.67 | 4.82 | Ref | |
| Medical assistant | 47 | 4.57 | 4.73 | 4.50 | −0.241 (−0.985 to 0.503) | 0.526 |
| Nurse/midwife | 105 | 4.62 | 4.63 | 4.62 | −0.149 (−0.834 to 0.536) | 0.669 |
| HSA | 190 | 4.13 | 4.11 | 4.13 | −0.820 (−1.500 to 0.142) | 0.018 |
| Hospital attendant | 172 | 3.13 | 3.04 | 3.19 | −1.605 (−2.286 to 0.923) | <0.001 |
| **Facility type** | | | | | | |
| Dispensary | 40 | 3.98 | 4.29 | 3.81 | Ref | |
| Health centre | 378 | 4.01 | 3.91 | 4.04 | 0.136 (−0.298 to 0.571) | 0.538 |
| Rural hospital | 113 | 3.80 | 3.76 | 3.92 | −0.004 (−0.501 to 0.493) | 0.987 |
| **District** | | | | | | |
| Mchinji | 193 | 3.87 | | | Ref | |
| Zomba | 338 | 4.01 | | | 0.001 (−0.264 to 0.266) | 0.993 |

Adjusted $R^2$: 0.173.

HSA, Health Surveillance Assistant; IMCI, Integrated Management of Childhood Illnesses.

Yes, it provides a holistic approach to childhood illness, it gives knowledge on how to treat under-five children. It is a lifesaving system. (Mchinji, FDG 1)

## IMCI sustainability

This theme addresses the long-term sustainability of IMCI implementation in frontline facilities, with both end-user ownership and sustaining capacity discussed. Notably, this topic was only raised by healthcare providers in Mchinji and included both challenges and concrete recommendations. A key challenge was around the transfer of trained staff, especially when not all staff are sent for IMCI training. Around who attends training, there were concerns around transparency of who is sent for training, which can compound the issue.

Yes, it is a problem, for instance someone has gone for IMCI training and next month has been transferred, that means we will have a problem with IMCI because that person didn't finish sharing the information. (Mchinji, FDG 2)

To promote sustainable programming there was a request for healthcare workers to be involved in the development process, and ensuring motivation and follow-up was sustained. NGO instituted programmes were specifically called out for this practice:

Another problem, it's you NGO people, you don't follow up the programmes. (Mchinji, FDG 3)

## DISCUSSION

In this study we explored how IMCI is currently being implemented within two districts in Malawi. We found that there were mixed experiences with both implementation and access to essential supplies, notably with important differences between target infections. In terms of healthcare workers, we found refresher training to be more important in terms of IMCI knowledge than cadre of the healthcare worker and having training alone. These findings pose important topics for further exploration and discussion.

The ability to diagnose and treat respiratory conditions in general, and particularly pneumonia, in frontline facilities was poor. This is especially notable when compared with the almost universal access to malaria RDTs and treatment. Oral rehydration salt, as a treatment for diarrhoeal diseases, was also not readily available at government facilities. The IMCI guidelines recommend seven essential drugs for treating malaria, pneumonia, diarrhoea and ear infections. Several facilities lacked the necessary equipment and first-line treatments to diagnose and treat pneumonia, with the exception of cotrimoxazole, which

is a first-line treatment for respiratory infections in children with HIV.

Donor funding for child health has steadily increased since the MDGs were introduced in 2000 and this has coincided with increased policies for malaria prevention and treatment in Malawi.[10] This is reflected in well-funded malaria programmes in Malawi (Malawi Malaria Communication Strategy 2015–2020[22]), and in the global funding landscape where malaria has been prioritised through the Global Fund, Bill and Melinda Gates Foundation and USAID.[23 24] This contrasts to pneumonia and diarrhoea, which despite having high burdens, do not receive the same level of strategic funding by large-scale donors. In particular, pneumonia has generally received lower research funding comparative to its global mortality burden.[25] This vertical programming focus is often disease specific and detracts from the holistic approach to assessing childhood illness, which IMCI promotes. Further exploration of factors which promote sustainability at different levels of the health system could reveal important insights for future programming.

Resource availability impacts healthcare workers' abilities to adequately perform IMCI assessments and treat according to guidelines. This was apparent from the FGDs with providers, and consistent with previous studies from similar settings, which found better health worker performance in facilities with availability of all essential IMCI equipment.[26] Given that some health facilities were better resourced than others, it would be useful to understand different funding allocation, stock management and personnel structures, to ascertain best practices.

The assessment of healthcare worker IMCI knowledge found more than half of healthcare workers scored under 50%, and those with higher qualifications did not necessarily score better. Completion of both IMCI training and a refresher course was associated with higher IMCI knowledge. This is in line with other studies on IMCI performance, where healthcare workers who had received IMCI training without a refresher course, performed similarly to those without training.[27] It is important to note that while training does not necessarily translate into knowledge, knowledge does not always result in adherence to IMCI guidelines and improved practice. A previous study found that those with more experience felt that following the guidelines was not necessary.[27] Time constraints and a high work load have also been reported to reduce adherence to IMCI guidelines.[28]

We found a weak positive relationship between time since training and knowledge, which may reflect the importance of long-term clinical experience rather than training alone. A systematic review by Rowe *et al* reported that healthcare workers performed better with IMCI training than without, but among those trained, over a third of ill children were not treated according to IMCI guidelines.[29] Traditional training programmes do not necessarily translate to adequate knowledge and skills without ongoing support, and findings from five sub-Saharan African countries found coaching and

mentorship improved IMCI service delivery.[30] The WHO recommends following up 4 weeks after the initial IMCI training course to help reinforce the skills learnt, and continued supervision.[31] There are examples of successful mentorship within child health,[32 33] and community case management in Malawi. However, as lacking supervision and mentorship was raised by healthcare workers in the FGDs, work is needed on effective adoption at frontline facilities.

The qualitative data highlighted a key issue around programme sustainability, and the need for local ownership. Training and refresher courses were often provided by NGOs (through external funding), not provided to all staff, and the decision for who is trained was not thought to be transparent. Given the frequent transfer of healthcare workers across different facilities, this results in an unequal distribution of workers with IMCI knowledge. It is therefore important that sustainability and handover plans are established by NGOs and research groups, which are agreed up-front with District Health Management Teams.

This study had three key limitations, first, that the two districts were selected purposively, and our findings may not be representative to other districts. For example, Mchinji has had multiple research projects in the last decade which have included IMCI refresher trainings and health system strengthening (eg, McCollum *et al*[34]). In particular, we did not include any district from the northern region. Second, the healthcare workers in the survey were those present on the day, and therefore may not cover the full range of providers (eg, those who predominantly focus on outreach services). The frequency of healthcare worker cadres generally reflected the health workforce composition in Malawi, where HSAs constitute one third of the workforce and few clinical officers.[19] The higher proportion of workers from Zomba is also expected given it has a larger population compared with Mchinji, and has traditionally received higher healthcare funding.[17] Finally, assessment of practice was not performed, with direct observations of health worker performance posing several ethical and methodological challenges (eg, the Hawthorne effect[35]). We therefore cannot comment on quality of IMCI care provision.

Malawi's early adoption and scaled-up implementation of IMCI has likely played a key role in the successes made in reducing under-five mortality, but further progress is needed to reach the SDG 3.2. While we found expected challenges in staffing and resources, there are several opportunities for improvement and further research. First, determining which combination of supervision, mentorship and retraining is most cost-effective within the specific context of frontline facilities, to ensure improved knowledge and subsequent quality of care. Second, to explore locally adapted IMCI best practices from those facilities which had good supply chain management, positive attitudes to staff transfers and sufficient trained staff capacity. Finally, supporting programming to move away from vertical financing with short term support, to

the more holistic approach which IMCI promotes and considers strategies for embedded sustainability.

**Author affiliations**
[1]Department of Global Public Health, Karolinska Institute, Stockholm, Sweden
[2]Astrid Lindgren Children's Hospital, Karolinska University Hospital, Stockholm, Sweden
[3]Department of Clinical Science, Intervention and Technology, Karolinska Institutet, Stockholm, Sweden
[4]Malawi Epidemiology and Intervention Research Unit (MEIRU), Lilongwe, Malawi
[5]Parent and Child Health Initiative, Lilongwe, Malawi
[6]Paediatrics, University of Malawi College of Medicine, Blantyre, Malawi
[7]Malawi-Liverpool-Wellcome Trust Programme, Blantyre, Malawi
[8]Acute Respiratory Infections Unit, Ministry of Health, Lilongwe, Malawi
[9]Institute for Global Health, University College London, London, UK

**Acknowledgements** We would like to thank the District Health Office in Mchinji and Zomba for supporting this project, and all the healthcare providers, facility managers, technicians and support start for sharing their time and knowledge. We acknowledge Christopher Kampinga, Tarcizious Phiri, Harry Liyaya (PACHI, Malawi) and Clifford Katumbi (College of Medicine, Malawi) for collecting the audit and survey data.

**Contributors** The study was conceived by HH and CK and study protocols and data collection tools designed together with BZ, CM, LB, ND and AD. Quantitative data collection was overseen by BZ, LB and CM and qualitative data collection was conducted by AD, with support from BZ. Quantitative analysis was conducted by KK with input from CK and HH. Qualitative analysis was conducted by CK and HH, with input from AD and KK. The manuscript was drafted by KK, with significant contributions from CK, HH and AD. All authors read and approved the final manuscript.

**Funding** This study was funded by a project grant the Swedish Research Council 2017-05579. The funder did not play any role in the design, analysis or interpretation of the data. The analysis and write up was undertaken as part of an MSc student project (the student received a Global Grant Scholarship through Rotary International to undertake a graduate study programme).

**Competing interests** None declared.

**Patient consent for publication** Not required.

**Provenance and peer review** Not commissioned; externally peer reviewed.

**Data availability statement** Data are available upon reasonable request. All relevant data reported in the paper is included in the tables. For raw data please contact Carina King (carina.king@ki.se) to request anonymised data for research purposes only, following approval from the College of Medicine Research Ethics Committee.

**ORCID iD**
Carina King http://orcid.org/0000-0002-6885-6716

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
