## [Reviewer comments · BMJ Paediatrics Open]

ARTICLE DETAILS

TITLE (PROVISIONAL)	Integrated Management of Childhood Illnesses (IMCI): A mixed-methods study on implementation, knowledge and resource availability in Malawi
AUTHORS	Kilov, Kim Hildenwall, Helena Dube, Albert Zadutsa, Beatiwel Banda, Lumbani Langton, Josephine Desmond, Nicola Lufesi, Norman Makwenda, Charles King, Carina

VERSION 1 – REVIEW

REVIEWER	Reviewer name: Dr. Eirini Koutoumanou Institution and Country: University College London, United Kingdom of Great Britain and Northern Ireland Competing interests: None
REVIEW RETURNED	23-Feb-2021

GENERAL COMMENTS	Thank you for the opportunity to comment on an interesting and well written report about the implementation of IMCI in two districts in Malawi. The authors have provided a clear description of where and how the data was collected but also acknowledged important limitations related to the study design. Please find below some recommendations (mainly minor) on several areas which, if addressed, could further improve this manuscript. I recommend that in the analysis section, the chi-square test is included as the means of comparison between proportions across different categories as this is only mentioned ones in the abstract. The counts shown in tables 2 and 3 are very interesting and relevant to the research question but are not presented separately for Zomba and Mchinji. Would it not be more relevant to split the counts by area? As I appreciate that this would make both tables look busier, the authors might want to consider downscaling the 'in-date' column as the majority was 100% - possibly mark only those that were not 100%? I recommend that the mean knowledge score and median time since training are presented with confidence intervals for a better sense of precision (either via non-parametric or bootstrapping techniques for the latter, as I suspect it was skewed as it is summarised via a median). I believe that the results presented in table 4 are from a single multiple linear regression model whose R squared value was 0.19.
---

	I would recommend reporting the adjusted R squared value. Also, did the authors explore the relevance and significance of interactions terms between the explanatory variables and the district variable? The authors mention in the discussion section that “The assessment of health care worker IMCI knowledge found more than half of healthcare workers scored under 50%...” – I suspect that this is derived directly from this study, but this result is not presented earlier in the results section. If the authors want to draw further conclusions from this, it might be worth, for better clarity, adding it to the results section too. Finally, one of the authors’ ultimate conclusions relates to moving away from vertical financing to a more holistic approach. However, I am not sure that the right kind of data and results have been presented in this paper to support this argument. The first mention of vertical funding comes in the discussion section without a clear enough definition of what it exactly stands for. If the authors believe that this is backed up from their analysis, could they make the transition from the data to the above statement clearer? Minor:  - In the abstract, edit ‘chi2 tests’ to ‘chi-square tests’ - Add a reference for the very first statement of this paper (opening line) - “...only 44 countries were considered to be fully implementing *it*.” and similarly for “Malawi was an early adopter of IMCI, first implementing *it* in 2000...” - Add column totals in Table 1 - Table 4 heading, replace the word multivariate with multivariable as per the text - Correct “work??? together effectively”
--	--

REVIEWER	Reviewer name: Dr. Elizabeth Mason Institution and Country: not applicable Competing interests: None
REVIEW RETURNED	02-Mar-2021

GENERAL COMMENTS	Overall: The article would benefit from an introduction on the different types of training for IMCI in Malawi, there has been significant investment in iCCM for the dispensary/community level, also information on whether IMCI is included in pre-service training. Setting: It would be useful to include a short description of the different cadres treating children at each level of Health Facility. Sampling: As purposeful sampling was used, including staff who interact with paediatric patients (46/47), please clarify if all the staff interviewed treat children, particularly the attendants at health centres/dispensaries, who are the least qualified and received no IMCI training. Quantitative data collection: IMCI drugs include pre-referral drugs, no mention of these is included, although there are a number of questions on referral. Is this data available? If so can it be included, or if not an explanation as to why not. IMCI practice uses job aides - the chart booklet. There is no mention of the chart booklet or its availability. Was this considered as a question and is this data available, if so it should be included. The knowledge questionnaires were adapted from USAID post course evaluation P7 12/13) yet the WHO/UNICEF guidelines on
---

	follow-up after training are referenced, as are the health facility surveys. An explanation of the choice of questionnaire would be useful. Results: Facility audits: Amoxicillin is available as tablets and syrup, it is not clear if both are available in the same facility or different facilities. Suggest grouping as a child can be given either. Health care worker survey: See previous comment regards IMCI and iCCM training, the latter was directed at HSA's. Given that only one of the attendants had no training, see previous question as to whether this cadre treats children. Section on mean knowledge scores P9 9-24, median would be more appropriate, with a bar chart of the range of scores. FGD: The discussion on sustainability does not include supportive supervision, was this question probed? also pre-service training. It is not clear when the last IMCI training was conducted, making this difficult to interpret. Discussion: P13 36-49. Donor funds have been increasing since MDGs were introduced, this is quite vague. Since 2000? and still increasing though MDGs finished in 2015? Please clarify. Suggest the section on HW knowledge be rewritten following the reanalysis of scores. Also it is well known that knowledge and practice do not necessarily correlate, there was no assessment of practice, this shortcoming should be explained. Although the final conclusion on moving from vertical programme funding to a more holistic approach is correct, it does not acknowledge the efforts made by Malawi on iCCM, it would be useful to discuss this re the community/dispensary level, vs the HC and hospital levels.
--	--

VERSION 1 – AUTHOR RESPONSE

Dear Prof. Choonara,

Thank you for sending this manuscript for review. The reviewers raised some helpful clarifications and the edits should provide a clearer manuscript. Please find the point by point response in the attached file.

Kind regards,

Dr Carina King, on behalf of the authors

Please note, all page and line numbers refer to the marked up version of the manuscript.

Reviewer: 1

Thank you for the opportunity to comment on an interesting and well written report about the implementation of IMCI in two districts in Malawi. The authors have provided a clear description of where and how the data was collected but also acknowledged important limitations related to the study design. Please find below some recommendations (mainly minor) on several areas which, if addressed, could further improve this manuscript.

We thank the reviewer for their kind review and constructive feedback.

1. I recommend that in the analysis section, the chi-square test is included as the means of comparison between proportions across different categories as this is only mentioned ones in the abstract.

- a. Thank you for noticing this missing information from the analysis section – we have now added this (pg. 6, line 31)*
2. The counts shown in tables 2 and 3 are very interesting and relevant to the research question but are not presented separately for Zomba and Mchinji. Would it not be more relevant to split the counts by area? As I appreciate that this would make both tables look busier, the authors might want to consider downscaling the ‘in-date’ column as the majority was 100% - possibly mark only those that were not 100%?
- a. We did look at presenting these data separately by district, but given the numbers of facilities was few, we felt it didn’t add as much information. However, the suggestion to remove the % in date, we agree and have done this. We are happy to add a disaggregated table, by district, to the appendix if the reviewer or editors would like.*
3. I recommend that the mean knowledge score and median time since training are presented with confidence intervals for a better sense of precision (either via non-parametric or bootstrapping techniques for the latter, as I suspect it was skewed as it is summarised via a median).
- a. We have now added these to the main results text (pg. 8, lines 6-7), and have presented the range instead of IQR to give a better sense of the data.*
4. I believe that the results presented in table 4 are from a single multiple linear regression model whose R squared value was 0.19. I would recommend reporting the adjusted R squared value. Also, did the authors explore the relevance and significance of interactions terms between the explanatory variables and the district variable?
- a. We have now reported the adjusted r-squared. In terms of interactions we did not originally look at these. We have now run models exploring interactions between district and facility type and cadre, and the main conclusion around refresher training remained unchanged. In both these new models the adjusted R-squared was lower than the model with no interaction terms, and the association with cadre remained – so given it didn’t materially change the interpretation, we prefer to retain the simpler model.*
5. The authors mention in the discussion section that “The assessment of health care worker IMCI knowledge found more than half of healthcare workers scored under 50%...” – I suspect that this is derived directly from this study, but this result is not presented earlier in the results section. If the authors want to draw further conclusions from this, it might be worth, for better clarity, adding it to the results section too.
- a. Thank you for noticing this oversight in our results reporting. We have now included a statement (pg. 8, line 10) and have added a figure to the supplementary appendix.*
6. Finally, one of the authors’ ultimate conclusions relates to moving away from vertical financing to a more holistic approach. However, I am not sure that the right kind of data and results have been presented in this paper to support this argument. The first mention of vertical funding comes in the discussion section without a clear enough definition of what it exactly stands for. If the

authors believe that this is backed up from their analysis, could they make the transition from the data to the above statement clearer?

a. Thank you for pointing this out – we have made edits throughout the discussion to try and make the point clearer. We saw clear differences in the resources available for different disease conditions, and these reflect the vertical national and international funding systems – namely for malaria. Given IMCI promotes a holistic approach to child health, its clear that funding drugs and diagnostics for individual conditions and not others is problematic. Additionally, as vertical programming (in this case disease specific programmes) is often funded externally, there are concerns for sustainability – which was raised by healthcare workers as a concern. We hope this makes our thinking clearer.

7. In the abstract, edit ‘chi2 tests’ to ‘chi-square tests’

a. This has been resolved (pg 2, line 11)

8. Add a reference for the very first statement of this paper (opening line)

a. This has been resolved (pg 4, line 2)

9. “...only 44 countries were considered to be fully implementing *it*.” and similarly for “Malawi was an early adopter of IMCI, first implementing *it* in 2000...”

a. This has been resolved (pg 4, line 12 & 23)

10. Add column totals in Table 1

a. This has been resolved

11. Table 4 heading, replace the word multivariate with multivariable as per the text

a. This has been resolved

12. Correct “work??? together effectively”

a. This has been resolved (pg 9, line 23)

Reviewer: 2

1. The article would benefit from an introduction on the different types of training for IMCI in Malawi, there has been significant investment in iCCM for the dispensary/community level, also information on whether IMCI is included in pre-service training.
 - a. *There is a mix of sources for IMCI training in Malawi, which is reflected in the responses given to the healthcare worker survey (result given on page 8) – we have expanded this to give a more detail. While some providers reported having IMCI training during their qualification, its apparent that this is inconsistent, and whether healthcare workers would have been exposed to this will depend on when they qualified. Given most training was provided by NGOs, there is likely to be variation in the content, pedagogical approach, and duration of trainings; however, the presence of district-level IMCI coordinators should mitigate some of these challenges.*
2. Setting: It would be useful to include a short description of the different cadres treating children at each level of Health Facility.
 - a. *We have now included a table in the appendix that provides a short description of some of the different cadres involved in treatment children. A statement addressing this has also been included in the text (pg. 5, line 19).*
3. Sampling: As purposeful sampling was used, including staff who interact with paediatric patients (46/47), please clarify if all the staff interviewed treat children, particularly the attendants at health centres/dispensaries, who are the least qualified and received no IMCI training.
 - a. *We decided to include facility staff that interact with children in a broad sense, and not restrict to facility staff that are designated or responsible for clinical assessments and treatment within those facilities (i.e. medical assistants and clinical officers). This was done to gain a wider picture of how IMCI is implemented, as we had informally observed HSAs and attendants take part in growth monitoring, triage, drug dispensing, health talks and general support when clinical staff are away.*
4. Quantitative data collection: IMCI drugs include pre-referral drugs, no mention of these is included, although there are a number of questions on referral. Is this data available? If so can it be included, or if not an explanation as to why not.
 - a. *In this study we decided to focus on routine case management, and not on emergency case management (note, a manuscript on ETAT preparedness from the same context is currently under-review, where we focus on the ability to deliver IV drugs and stabilisation of critically ill children). Given pre-referral treatment under IMCI includes giving the first dose of amoxicillin, locally recommended anti-malarial, starting ORS treatment, and paracetamol – depending on the diagnosis. All of these are covered by the data reported. We do appreciate however, that there is a lot more treatment options available when managing a critically ill child, and have added a statement in the Methods to clarify this was not our focus (pg 6, line 7).*
5. IMCI practice uses job aides - the chart booklet. There is no mention of the chart booklet or its availability. Was this considered as a question and is this data available, if so it should be included.
 - a. *Thank you for raising this, and it was an oversight on our part when designing the study to not collect data on the availability of job aides, use of reporting tools and M&E processes.*
6. The knowledge questionnaires were adapted from USAID post course evaluation P7 12/13) yet the WHO/UNICEF guidelines on follow-up after training are referenced, as are the health facility surveys. An explanation of the choice of questionnaire would be useful.
 - a. *The questionnaire was chosen as it provided a set of short and comparable questions that had been used in a similar setting. We did not want to use any knowledge questions that were included in any of the routine IMCI training tools, or adapt any of the training*

scenarios in case this biased respondents who had seen them before. We have added a short statement in the Methods to clarify this (pg 6, line 10).

7. Results: Facility audits: Amoxicillin is available as tablets and syrup, it is not clear if both are available in the same facility or different facilities. Suggest grouping as a child can be given either.
 - a. *We decided to present tablets and syrup separately, based on feedback during the field pilot, where recording stocks and the volume of drugs in date was hard to combine. As the formulation of drugs can be important for adherence and different dosing/formulations are recommended for difference age groups, we thought it interesting to keep these separate. However, we take the point that facilities might have one and not the other, or vice versa, and so have added information on this into the footnote of Table 2.*

8. Health care worker survey: See previous comment regards IMCI and iCCM training, the latter was directed at HSA's. Given that only one of the attendants had no training, see previous question as to whether this cadre treats children.
 - a. *Reflecting, it was an oversight for us not to have asked about other forms of training that healthcare providers had received, including iCCM training. In this sample, HSAs were the mostly likely cadre to report having had IMCI training, they still scored worse than higher cadres of healthcare workers. And conversely, attendants who had not been trained still scored a mean of 3.13/10, suggesting some knowledge had been acquired through practice or local training dissemination. However, we fully acknowledge that not being able to check if providers had iCCM training is a limitation.*

9. Section on mean knowledge scores P9 9-24, median would be more appropriate, with a bar chart of the range of scores.
 - a. *We have now included a supplementary figure showing the range of scores. We chose to present the mean, and not the median given the data are normally distributed and the mean and median are very similar (3.96 and 4, respectively).*

10. FGD: The discussion on sustainability does not include supportive supervision, was this question probed? also pre-service training. It is not clear when the last IMCI training was conducted, making this difficult to interpret.
 - a. *We did not ask questions directly about supportive supervision and sustainability was something spontaneously raised by some participants, rather than something we were specifically probing about. As we did not include identifying questions in the knowledge survey, we were not able to link the data collected in the FGDs with the time since training for individual participants.*

11. Discussion: P13 36-49. Donor funds have been increasing since MDGs were introduced, this is quite vague. Since 2000? and still increasing though MDGs finished in 2015? Please clarify.
 - a. *Malaria programmes have been well funded in Malawi in the past, with donor funding from a number of sources including USAID through the US President's Malaria initiative and the Malawi Malaria Communication Strategy 2015-2020. We have now included an additional reference to the Malawi Malaria Communication strategy which highlights one of the programs that has continued since the MDGs finished in 2015*

(pg 11, line 26).

12. Suggest the section on HW knowledge be rewritten following the reanalysis of scores. Also it is well known that knowledge and practice do not necessarily correlate, there was no assessment of practice, this shortcoming should be explained.

- a. Thank you for raising this. We agree that this is a limitation to the study and have now included this as a limitation in the discussion (Pg 13, line 16); however, we have kept the IMCI knowledge scores presented and analysed as means (see response above).*

Although the final conclusion on moving from vertical programme funding to a more holistic approach is correct, it does not acknowledge the efforts made by Malawi on iCCM, it would be useful to discuss this re the community/dispensary level, vs the HC and hospital levels.

- b. Thank you for this comment. We agree that iCCM has also played a significant role; however, addressing this is beyond this scope of this particular study and we did not ask questions about iCCM practice. We have added a short statement that highlights the importance of exploring other factors that promote sustainability at different levels of the health care system (Pg. 11, line 31).*